# SELF-DISTILLED DISENTANGLEMENT FOR COUNTER-FACTUAL PREDICTION

## ABSTRACT

The advancements in disentangled representation learning significantly enhance the accuracy of counterfactual predictions by granting precise control over instrumental variables (IVs), confounders, and adjustable variables. An appealing method for achieving the independent separation of these factors is mutual information minimization (MIM), a task that presents challenges in numerous machine learning scenarios, especially within high-dimensional spaces. To circumvent this challenge, a common strategy is to re-frame the MIM problem from a problem between two high-dimensional representations to one between high-dimensional representations and low-dimensional labels based on the different dependencies of latent factors and known labels. In this paper, we first demonstrate the limitations of this approach in separating instrumental variables and confounding variables, as determined by the d-separation theory. Subsequently, we propose the Self-Distilled Disentanglement framework, referred to as $SD^2$. Grounded in information theory, it ensures theoretically sound disentangled representations without intricate mutual information estimator designs for high-dimensional representations. Our comprehensive experiments, conducted on both synthetic and real-world datasets, provide compelling evidence of the effectiveness of our approach in facilitating counterfactual inference in the presence of both observed and unobserved confounders.

## 1 INTRODUCTION

Counterfactual prediction has attracted increasing attention (Alaa & van der Schaar, 2017; Li et al., 2016; Chernozhukov et al., 2013; Glass et al., 2013) in recent years due to the rising demands for robust and trustworthy artificial intelligence. Confounders, the common causes of treatments and effects, induce spurious relations between different variables, consequently undermining the distillation of causal relations from associations. Thanks to the rapid development of representation learning, a plethora of methods (Li & Yao, 2022; Yao et al., 2018; Shalit et al., 2017) mitigate the bias caused by the observed confounders by generating balanced representations in the latent space. As for the bias brought by the unobserved confounders, Pearl et al. (2000); Angrist & Imbens (1994); Hartford et al. (2017); Muandet et al. (2020); Lin et al. (2019) propose obtaining an unbiased estimator by regressing outcomes on Instrumental variables (IVs), which are exogenous variables related to treatment and only affect outcomes indirectly via treatment, to break the information flow between unobserved confounders and the treatments.

There are two drawbacks to the above methods. Firstly, most of them treat all observed features as the observed confounders to block, while only part contributes to the distribution discrepancy of pre-treatment features. Secondly, the prerequisite for IV-based regression methods is to access valid IVs, which have three strict conditions (**Relevance**, **Exclusion**, **Unconfoundeness** (Wu et al., 2022a)) to satisfy, making it a thorny task to find.

Disentangled representation learning (Hassanpour & Greiner, 2019; Kingma & Welling, 2013), aiming at decomposing the representations of different underlying factors from the observed features, is showing promise in addressing the flaws above simultaneously. The disentangled factors provide us with a more precise inference route to alleviate the bias brought by the observed confounders. Additionally, one can automatically obtain the representations of the valid IVs to remove the bias led by the unobserved confounders.

Minimizing the mutual information (MI) between the representations of distinct underlying factors is fundamental to obtaining disentangled representations for counterfactual prediction (Cheng et al., 2022b). Nevertheless, estimating MI between high-dimensional representations remains a persistent challenge (Belghazi et al., 2018; Poole et al., 2019). Hassanpour & Greiner (2019); Yuan et al. (2022) leverage the different dependencies between underlying factors and known labels to avoid complex MI estimation between high-dimensional representations. However, these approaches lack a solid theoretical assurance for distinguishing IVs from confounders.

To overcome the defects of previous methods, we propose a novel algorithm for counterfactual prediction named Self-Distilled Disentanglement ($SD^2$) to sidestep MI estimation between high-dimensional representations from an alternative perspective. Specifically, we provide theoretical analysis rooted in d-separation and information theory to guarantee the disentanglement of diverse underlying factors. Further, we give a solvable form of our method through rigorous mathematical derivation to minimize MI directly rather than explicitly estimating it. Based on the theory put forward, we design a hierarchical distillation framework to kill three birds with one stone: disentangle three different underlying factors, mitigate the confounding bias, and grasp sufficient supervision information for counterfactual prediction.

Our main contributions are summarized as follows:

- We justify the infeasibility of separating instrumental variables and confounders if solely through the different dependence with known variables. Instead, we provide a theoretically assured solution rooted in d-separation and information theory for disentanglement to avoid mutual information estimation for high-dimensional representations.

- We provide a tractable form of our proposed solution through mathematically rigorous derivation, which rewrites the loss function of disentanglement and fundamentally tackles the difficulty of mutual information estimation.

- We propose a novel self-distilled disentanglement method ($SD^2$) for counterfactual prediction. By designing a hierarchical distillation framework, we disentangle IVs and confounders from observational data to mitigate the bias induced by the observed and unobserved confounders at the same time.

- We conduct extensive experiments on synthetic and real-world benchmarks to verify our theoretically grounded strategies. The results demonstrate the effectiveness of our framework on counterfactual prediction compared with the state-of-the-art baselines.

## 2 RELATED WORK

### 2.1 COUNTERFACTUAL PREDICTION

The main challenge for counterfactual prediction is the existence of confounders. Researchers adopt matching (Rosenbaum & Rubin, 1983), re-weighting (Imbens, 2004), regression (Chipman et al., 2010), representation learning (Shalit et al., 2017) to alleviate the confounding bias under the "no hidden confounders" assumption. To relax this unpractical assumption, a few non-parametric or semi-parametric methods utilize special structures among the variables to resolve the bias led by the unobserved confounders. These structures include (1) proxy variables (Veitch et al., 2019; 2020; Louizos et al., 2017; Wu & Fukumizu, 2022); (2) multiple causes (Wang & Blei, 2019; Zhang et al., 2019; Cheng et al., 2022a); (3) instrumental variables. 2SLS (Angrist & Imbens, 1994) is the classical IV method in a linear setting. Singh et al. (2019); Wu et al. (2022a); Xu et al. (2020); Hartford et al. (2017); Lin et al. (2019) adopt advanced machine learning or deep learning algorithms for non-linear scenarios. Another commonly used causal effects estimator using IVs is the control function estimator (CFN) (Wooldridge, 2015; Puli & Ranganath, 2020). These methods require well-predefined IVs or assume all observed features as confounders, which inevitably impairs their generalization to real-world practice. Yuan et al. (2022); Wu et al. (2022b) aims to generate IVs for downstream IV-based methods rather than focusing on counterfactual prediction.

## 2.2 DISENTANGLED REPRESENTATION LEARNING

Most of the current state-of-the-art disentangled representation learning methods are based on Kingma & Welling (2013), which uses a variational approximation posterior for the inference process of the latent variables. Typical work includes $\beta$-VAE (Higgins et al., 2017), Factor-VAE (Kim & Mnih, 2018), CEVAE (Louizos et al., 2017) and so on. Another popular solution (Chen et al., 2016; Zou et al., 2020) builds on the basis of Generative Adversarial Networks (Goodfellow et al., 2014). However, these studies are more suitable for the approximate data generation problem and fall short when it comes to estimating causal effects, largely due to the difficulty in training complex generation models. As the explicit identification of the underlying factors in observed features helps to alleviate confounding bias and improve inference accuracy, DRCFR (Hassanpour & Greiner, 2019) first introduces disentanglement methods into the field of counterfactual regression. DeR-CFR (Wu et al., 2020) designs some decomposition regularizers to ensure the separation of IVs and confounders. Despite previous efforts, existing representation methods are presented under the unconfoundedness assumption, which hinders their widespread generalization.

Compared with previous counterfactual prediction methods, $SD^2$ is the early pioneer of disentangled representation work equipped with a rigorous theoretical derivation that removes unobserved and observed confounding bias simultaneously. Our approach abandons well-predefined IVs but rather decomposes them from pre-treatment variables. Besides, we only impose conditions on variables that are related to confounding bias. Compared with generative disentangled methods, rather than generating latent variables, $SD^2$ directly disentangles the observed features into three underlying factors by introducing causal mechanisms, which is more efficient and effective.

## 3 PROBLEM SETUP

As shown in Figure 1(a), we have observed pre-treatment features $X$, treatment $T$, and outcome $Y$. $X$ is composed of three types of underlying factors: **Instrumental variable** $Z$ that only directly influences $T$. **Confounder** $C$ that directly influences both $T$ and $Y$. **Adjustable variable** $A$ that only directly influences $Y$. Besides these, some **Unobserved confounders** $U$ impede the counterfactual prediction. $Z, C, A, U$ in this paper are exogenous variables.

**Identification:** As stated in (Hartwig et al., 2023), suppose the generated IVs satisfy three conditions of valid IVs, additional assumptions are necessary to identify the average causal effect (ACE) of T on Y. Adequate assumptions encompass homogeneity in the causal impact of T on Y, uniformity in the relationship between Z and T, and the absence of effect modification.

We aim first to disentangle $Z$, $C$, and $A$ from $X$ based on the theoretical analysis, then propose a unified framework to tackle confounding bias caused by $C$ and $U$ simultaneously. Intuitive thought is to minimize the mutual information between the representation of $Z$, $C$ and $A$ during training. However, it is hard due to the notorious difficulty in estimating mutual information in high dimensions, especially for latent embedding optimization (Belghazi et al., 2018; Poole et al., 2019).

To circumvent this challenge, Hassanpour & Greiner (2019); Yuan et al. (2022) attempt to disentangle underlying factors according to the different dependencies between these factors and known variables (typically low-dimensional, e.g., $T$ and $Y$). In this way, they can reduce the complexity of the problem by re-framing it as an estimation of MI between a high-dimensional representation and a known label instead of between two high-dimensional representations. Hassanpour & Greiner (2019) proposes to disentangle $A$ from $X$ based on the following proposition.

**Proposition 3.1.** *Adjustment variable $A$ should be independent of treatment $T$.*

Notice that only $A$ is independent of $T$ while $Z$ and $C$ are not, which justifies the separation of $A$ from $X$. Inspired by Hassanpour & Greiner (2019), Yuan et al. (2022) assumes $Z$ being independent of $Y$ when conditioning on $T$. However, this would lead to a contradictory conclusion, as the open of the collider structure $Z \rightarrow T \leftarrow C$ will enforce the reliance of $Z$ on $C$ once $T$ is fixed as a condition, which consequently results in the dependence of $Z$ and $Y$ through $C$. Based on the d-separation (Pearl et al., 2000), we propose the following proposition.

**Proposition 3.2.** *Instrumental variable $Z$ and confounder $C$ have the same dependence with treatment $T$, outcome $Y$ and adjustable variable $A$.*

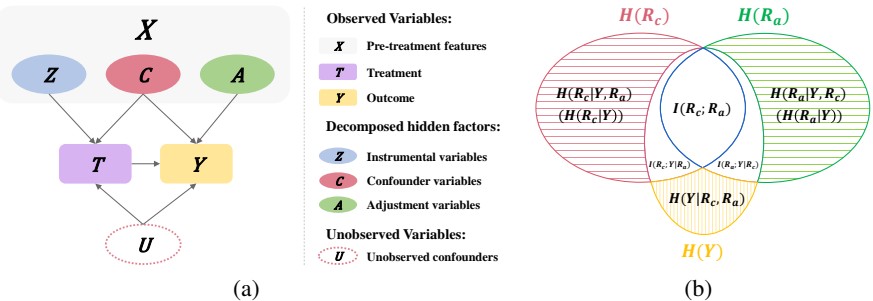

(a)                                     (b)

Figure 1: (a) General causal structure. Causal disentangled learning aims to automatically decompose $Z$, $C$, and $A$ from observed features $X$. (b) A motivating Venn diagram of mutual information between the representations of $C$, $A$, i.e., $R_c$, $R_a$ and outcome $Y$ during the training phase.

Detailed proof can be found in the Appendix A.1.1. **Proposition 3.2** indicates that it is infeasible to separate $Z$ and $C$ solely based on the different (in)dependent conditions with other known variables. Fortunately, we have discovered an alternative way to separate $Z$ and $C$ and bypass MI estimation between high-dimensional representations from the perspective of information theory, which will be demonstrated in Section 4. For the simple exposition, we display our theory and framework under the binary treatment and outcome setting, while the details of continuous setting are deferred to the Appendix A.3.2.

## 4 METHODOLOGY

### 4.1 THEORETICAL FOUNDATION OF SELF-DISTILLED DISENTANGLEMENT

**Disentangling $A$:** We define the representations of $Z$, $C$, and $A$ as $R_z$, $R_c$, and $R_a$ respectively. According to **Proposition 3.1**, $A \perp T$. Thus, we can decompose $A$ from $X$ by minimizing the distribution discrepancy of representations of $A$ between the treatment group ($T = 1$) and the control group ($T = 0$). The related loss function can be defined as follows:

$$\min \mathcal{L}_a = disc(\{R_a^i\}_{i:t_i=0}, \{R_a^i\}_{i:t_i=1}), \tag{1}$$

where function $disc(\cdot)$ represents the distribution discrepancy of $R_a$ between the treatment and control groups while i refers to the i-th individual.

**Disentangling $Z$:** Let's revisit the causal structure depicted in Figure 1(a). We notice that when we utilize $T$ as a feature to predict $Y$, including information from $Z$ becomes redundant, which motivates us to formulate the following objective function to ensure that the generated representation of $Z$ aligns with this observation:

$$\min \mathcal{L}_z = D_{KL}[\mathcal{P}_Y^T \| \mathcal{P}_Y^{R_z,T}], \tag{2}$$

**Disentangling $C$:** Given the obtained representations of both $A$ and $Z$, we can further disentangle $R_a$ from $X$. Observing the collider structures $C \rightarrow Y \leftarrow A$ and $C \rightarrow T \leftarrow Z$ in Figure 1(a), we notice that $C$ is independent of both $A$ and $Z$. This suggests that our objective should involve simultaneously minimizing both $I(R_a; R_c)$ and $I(R_z; R_c)$.

To circumvent the challenge of estimating mutual information between high-dimensional representations during training, we decompose the mutual information between $R_a$ and $R_c$ by leveraging the chain rule of mutual information:

$$I(R_a; R_c) = I(Y; R_c) + I(R_a; R_c \mid Y) - I(R_c; Y \mid R_a). \tag{3}$$

We further inspect the term $I(R_a; R_c \mid Y)$ in equation 3. Based on the definition of mutual information, we have,

$$I(R_a; R_c \mid Y) = H(R_a \mid Y) - H(R_a \mid Y, R_c), \tag{4}$$

where $H(\cdot)$ represents the function of entropy. Intuitively, this conditional mutual information measures the information contained in $R_a$ that is related to $R_c$ but unrelated to $Y$. As shown in Figure

1(a), the only connection between $C$ and $A$ is that they determine $Y$ jointly with $T$. If we set up two prediction models from $R_a$ and $R_c$ to Y, respectively, then the mutual information between $R_a$ and $R_c$ is all related to $Y$ during the training phase. We draw the Venn diagram of mutual information between $R_a$, $R_c$ and $Y$ as shown in Figure 1(b), according to which we have,

$$H(R_a \mid Y) = H(R_a \mid Y, R_c). \tag{5}$$

With equation 3, equation 4 and equation 5, we have,

$$I(R_a; R_c) = I(Y; R_c) - I(R_c; Y \mid R_a). \tag{6}$$

equation 6 transforms the mutual information between two training high-dimensional representations into the subtraction of two mutual information estimators with known labels. To further simplify it, we introduce the following theory:

**Theorem 4.1.** *Minimizing the mutual information between $R_a$ and $R_c$ is equivalent to:*

$$\min H(Y) - H(Y \mid R_c) - H(Y \mid R_a) + H(Y \mid R_c, R_a). \tag{7}$$

To find a tractable solution to equation 7, inspired by Tian et al. (2021), we derive the following Corollary:

**Corollary 4.2.** *One of sufficient conditions of minimizing $I(R_a; R_c)$ is minimizing the following conditions together:*

$$\min \mathcal{L}_c^a = \begin{cases} D_{KL}[\mathcal{P}_Y \| \mathcal{P}_Y^{R_a}] & \text{(8a)} \\ D_{KL}[\mathcal{P}_Y^{R_a} \| \mathcal{P}_Y^{R_c}], & \text{(8b)} \end{cases}$$

*where $\mathcal{P}_Y^{R_a} = p(Y \mid R_a)$, $\mathcal{P}_Y^{R_c} = p(Y \mid R_c)$ represent the predicted distributions, $\mathcal{P}_Y = p(Y)$ represents the real distribution.*

Detailed proof and formal assertions of **Theo.4.1** and **Corol.4.2** can be found in the Appendix A.1.2 and A.1.3. Similarly, we can transform minimizing $I(R_z; R_c)$ into following:

$$\min \mathcal{L}_c^z = \begin{cases} D_{KL}[\mathcal{P}_T \| \mathcal{P}_T^{R_z}] & \text{(9a)} \\ D_{KL}[\mathcal{P}_T^{R_z} \| \mathcal{P}_T^{R_c}], & \text{(9b)} \end{cases}$$

where $\mathcal{P}_T^{R_a} = p(T \mid R_a)$, $\mathcal{P}_T^{R_z} = p(T \mid R_z)$ represent the predicted distributions, $\mathcal{P}_T = p(T)$ represents the real distribution.

## 4.2 SELF-DISTILLED DISENTANGLEMENT FRAMEWORK

With the theory put forward in section 4.1, we propose a self-distilled framework to disentangle different underlying factors. To clarify further, we take the distillation unit for minimizing $\mathcal{L}_c^z$ to illustrate how we employ different sources of supervision information to directly minimize mutual information without explicitly estimating it.

As shown in Figure 2, **Retain** network represents the neural network for retaining the information from both Z and C. **Deep** networks and **Shallow** networks are named based on their relative proximity to $R_a$ and $R_c$. We set up two shallow prediction networks from $R_z$ and $R_c$ to $T$, respectively. In addition, to ensure $R_z$ and $R_c$ grasp sufficient information for predicting $T$, we set up a Retain network to concatenate $R_z$ and $R_c$ and store joint information of them for input into a deep prediction network. To minimize $\mathcal{L}_c^z$, we deploy distinct sources of supervision information from:

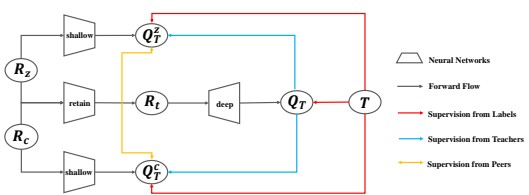

Figure 2: Self-distillation unit for minimizing $\mathcal{L}_c^z$.

(1) **Labels;** We use $T$ directly to guide the training of deep prediction networks by minimizing the cross-entropy (CE) loss $L(Q_T, T)$. For shallow prediction networks, we reduce $D_{KL}[\mathcal{P}_T \| \mathcal{P}_T^{R_z}]$ and $D_{KL}[\mathcal{P}_T \| \mathcal{P}_T^{R_c}]$ by minimizing CE loss $L(Q_T^z, T)$ and $L(Q_T^c, T)$.[1]

---

[1]In practice, only minimizing equation 9a and equation 9b makes convergence challenging, resulting in unsatisfactory performance. We speculate that this is due to a lack of sufficient supervised information guiding the updating direction of the prediction network for $R_c$. Therefore, we introduce the minimization of $D_{KL}[\mathcal{P}_T \| \mathcal{P}_T^{R_c}]$ (corresponding loss function $L(Q_T^c, T)$ to expedite loss convergence.

(2) **Teachers;** We regard retain and deep prediction networks as a teacher model, which can convey the learned knowledge to help the training of shallow prediction networks. That is, minimizing KL-divergence loss $L(Q_T^z, Q_T)$ and $L(Q_T^c, Q_T)$.

(3) **Peers;** We diminish the KL-divergence between the distributions of the outputs from the two shallow prediction networks, i.e., $L(Q_T^c, Q_T^z)$. $D_{KL}[\mathcal{P}_T^{R_z} \| \mathcal{P}_T^{R_c}]$ are consequently minimized.

Similarly, we can minimize $\mathcal{L}_z$ and $\mathcal{L}_c^a$ by establishing corresponding distillation units according to equation 2 and **Corol.4.2**. Combining all the self-distillation units, we have our total **S**elf-**D**istilled **D**isentanglement framework, i.e., $SD^2$.

**Mitigating Confounding Bias:** With disentangled confounders $C$, **the confounding bias induced by** $C$ can be alleviated by re-weighting the factual loss $L(Q_Y, Y)$ with the context-aware importance sampling weights $\omega_i$ (Hassanpour & Greiner, 2019) for each individual $i$. Besides, $Q_T$, the output of the deep prediction network for $T$, can be employed to regress $Y$, which helps to mitigate **the confounding bias caused by unobserved confounders** $U$ (Hartford et al., 2017).

The loss function of $SD^2$ is thus devised as follows with adjustable hyper-parameters $\alpha, \beta, \gamma, \delta$ and model parameters $W$.

$$\mathcal{L}_{SD^2} = \underbrace{\sum_i \omega_i L(Q_{Y_i}, Y_i) + \alpha L(Q_T, T)}_{factual\ loss} + \underbrace{\beta \mathcal{L}_a + \gamma(\mathcal{L}_z + \mathcal{L}_c^a + \mathcal{L}_c^z)}_{disentanglement\ loss} + \underbrace{\delta \|W\|_2}_{regularization\ loss} . \quad (10)$$

## 5 EXPERIMENTS

### 5.1 BENCHMARKS

Due to the absence of counterfactual outcomes in reality, it is challenging to conduct counterfactual prediction on real-world datasets. Previous works (Li & Yao, 2022) synthesize datasets or transform real-world datasets. In this paper, we have conducted multiple experiments on a series of synthetic datasets and a real-world dataset, Twins.

**Simulated Datasets**: *Binary Scenario*: The *synthetic* datasets are generated with the following steps: (1) For $K$ in $Z, C, A, V, U$, sample $K$ from $\mathcal{N}(0, I_{m_k})$, where $I_{m_k}$ denotes $m_k$ degree identity matrix. Concatenate $Z$, $C$, and $A$ to constitute the observed covariates matrix $X$. $V$ represents the observed IVs, the dimension of which could be set to 0. The setting of $V$ is mainly for the comparison of IV-based methods. $U$ represents the unobserved confounders. (2) Sample $T$ and $Y$ with Bernoulli distribution using $Z, C, V, U$ and $A, C, U$ as features, respectively. See the Appendix A.3.1 for the specific sampling equations.

*Continuous Scenario*: Following the work of Hartford et al. (2017); Wu et al. (2022a), we use the *Demand* datasets to evaluate the performance of $SD^2$ on the continuous scenario with the same data generation process described in detail in Wu et al. (2022a).

**Real-World Datasets** *Twins*: Detailed descriptions can be referred in (Wu et al., 2022a). Since only part of the features determines the outcome, we randomly generate $m_v$-dimension $V$ and choose some features $M$ as a combination of $Z$, $C$ and $U$ to create $T$ with the same policy in synthetic datasets. The rest features $R$ will include $A$ and some noise naturally. We hide some features in $M$ during training to create $U$ and treat the rest in $M$ with $R$ as $X$. During training, only $X$, $V$, $T$ and $Y$ will be accessible.

### 5.2 COMPARISON METHODS AND METRICS

The **baselines** can be classified into two categories: (1) **IV-based Methods**: DeepIV-Log and DeepIV-Gmm (Hartford et al., 2017), DFIV (Xu et al., 2020), OneSIV (Lin et al., 2019), CBIV (Wu et al., 2022a). These methods need well-predefined IVs $V$. When there is no $V$, i.e., $m_v$ equals 0, we use $X$ as $V$ for comparison. (2) **Non-IV-based Methods**: DirectRep, CFR-Wass (Shalit et al., 2017), DFL (Xu et al., 2020), DRCFR (Hassanpour & Greiner, 2019), CEVAE (Louizos et al., 2017). The latter two also disentangle the hidden/latent factors from observed features.

Table 1: Performance comparison of bias of ATE between $SD^2$ and the SOTA baselines on the synthetic datasets ($m_v$-$m_z$-$m_c$-$m_a$-$m_u$) and Twins datasets (Twins-$m_v$-$m_x$-$m_u$) in out-of-sample setting. Bold/Underline indicates the method with the best/second-best performance.

| Out-of-Sample | Synthetic | | | | Twins | | | |
|---|---|---|---|---|---|---|---|---|
| Method | 0-4-4-2-2 | 2-4-4-2-2 | 0-4-4-2-10 | 0-6-2-2-2 | Twins-0-16-8 | Twins-4-16-8 | Twins-0-16-12 | Twins-0-20-8 |
| DirectRep | 0.034(0.024) | 0.034(0.013) | 0.060(0.015) | 0.037(0.024) | 0.023(0.011) | 0.014(0.013) | 0.023(0.016) | 0.014(0.010) |
| CFR | 0.027(0.016) | 0.032(0.021) | 0.031(0.024) | 0.054(0.018) | 0.016(0.008) | 0.019(0.010) | 0.025(0.019) | 0.019(0.014) |
| DFL | 6.796(0.056) | 6.750(0.086) | 7.398(0.051) | 6.393(0.063) | 0.243(0.136) | 0.252(0.140) | 0.240(0.141) | 0.245(0.145) |
| DRCFR | 0.053(0.016) | 0.050(0.023) | 0.071(0.020) | 0.059(0.018) | 0.020(0.019) | 0.018(0.007) | 0.020(0.009) | 0.028(0.018) |
| DEVAE | 0.420(0.009) | 0.432(0.008) | 0.372(0.005) | 0.456(0.008) | 0.043(0.009) | 0.039(0.008) | 0.038(0.008) | 0.034(0.008) |
| DeepIV-Log | 0.572(0.028) | 0.567(0.016) | 0.601(0.011) | 0.537(0.021) | 0.028(0.016) | 0.026(0.019) | 0.028(0.018) | 0.018(0.011) |
| DeepIV-Gmm | 0.469(0.008) | 0.387(0.021) | 0.517(0.009) | 0.442(0.004) | 0.017(0.011) | 0.013(0.009) | 0.017(0.011) | 0.013(0.011) |
| DFIV | 7.047(0.131) | 6.856(0.093) | 7.749(0.125) | 6.585(0.130) | 0.257(0.145) | 0.255(0.148) | 0.256(0.144) | 0.227(0.151) |
| OneSIV | 0.507(0.009) | 0.441(0.064) | 0.569(0.015) | 0.484(0.014) | 0.016(0.014) | 0.011(0.007) | 0.025(0.020) | 0.013(0.010) |
| CBIV | 0.059(0.024) | 0.067(0.030) | 0.049(0.023) | 0.030(0.017) | 0.033(0.016) | 0.063(0.017) | 0.057(0.039) | 0.043(0.022) |
| Ours | **0.012(0.008)** | **0.022(0.017)** | **0.029(0.018)** | **0.013(0.010)** | **0.012(0.007)** | **0.011(0.006)** | **0.012(0.006)** | **0.011(0.009)** |

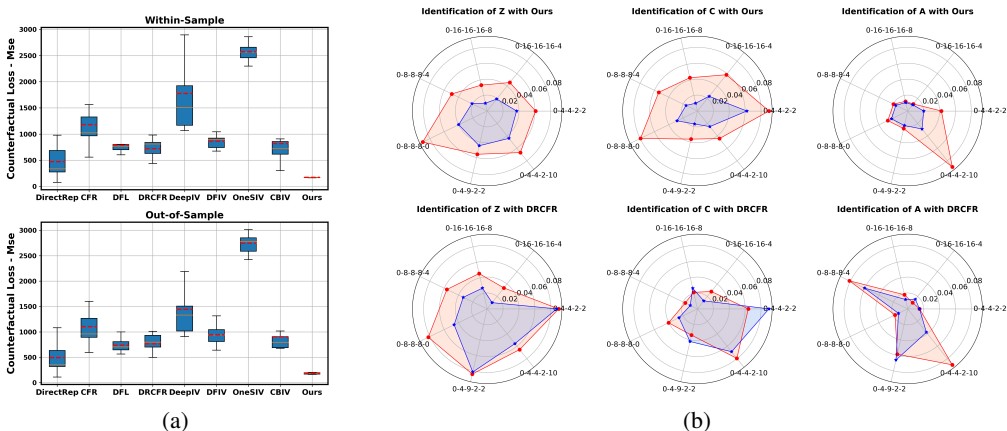

(a)                   (b)

Figure 3: (a) Experimental results under continuous scenario on Demand-0-1. (b) Radar charts visualizing the capability of $SD^2$ and DRCFR. The red and blue denote the contribution of actual and other variables to the decomposed representations.

**Metrics**: We use the absolute bias of Average Treatment Effect, i.e., $\epsilon ATE$, to evaluate the performance of different algorithms in the binary scenario. Formally,

$$\epsilon ATE = |\ \frac{1}{N}[\sum_{i=1}^{N}(Y_i^1 - Y_i^0) - \sum_{i=1}^{N}(\hat{Y_i^1} - \hat{Y_i^0})]\ |, \tag{11}$$

where $Y_i/\hat{Y_i}$ represents the factual/predicted potential outcome. For continuous scenarios, we take Mean Squared Error ($MSE$) as the evaluation metric. **The smaller $\epsilon ATE$ and $MSE$ are, the better the performance.**

## 5.3 RESULTS

### 5.3.1 COMPARISON WITH THE SOTA METHODS

**Binary Scenario**: Table 1 shows the performance of $SD^2$ on datasets with binary values. $m_v$-$m_z$-$m_c$-$m_a$-$m_u$ represents the data generated with $m_v$ predefined IVs $V$, $m_z$ underlying IVs $Z$, $m_c$ underlying confounders $C$, $m_a$ underlying adjustable variables $A$ and $m_u$ unobserved confounders $U$. During training, we can only observe $Z$, $C$ and $A$ as a whole $X$. Twins-$m_v$-$m_x$-$m_u$ denotes the

Twins dataset with $m_v$ predefined IVs, $m_x$ observed pre-treatment covariates and $m_u$ unobserved confounders and IVs. For data setting with $m_x = 16$ (Twins-0-16-8, Twins-4-16-8, Twins-0-16-12), there are 12 confounders and IVs, 4 adjustable variables and noises, while for Twins-0-20-8, we add 4 confounders and IVs in $X$. We generate 10000 samples for synthetic datasets for training/validation/testing sets, respectively, For the Twins datasets, we randomly choose 5271 samples and split the datasets with the ratio 63/27/10. We perform 10 replications and report the mean and standard deviation of the bias of ATE estimation. The results of the out-of-sample cases are presented in the Table 1 while **we leave the results of the within-sample cases in the Appendix A.2.1**.

Taking the results in 0-4-4-2-2 as a base point for observation, we have the following findings:

- The results of almost all IV-based methods in 0-4-4-2-2 are worse than those in 2-4-4-2-2, demonstrating the performance of IV-based methods is highly dependent on the predefined instrumental variables.

- By comparing the results between 0-4-4-2-2 and 0-4-4-2-10, we find that when there are more unobserved confounders in the dataset, the performance of confounder balancing methods (such as CFR, DRCFR, and DFL) will be poorer, which indicates the necessity of controlling the unobserved confounders.

- If there are fewer confounders in observed features, as we compare 0-4-4-2-2 with 0-2-6-2-2, the performance of confounder balancing methods without disentanglement (such as CFR, DFL) gets impeded, implying the significance of precise confounder control by decomposing underlying factors.

- $SD^2$ achieves the best performance among all data settings and is far better than the second-best methods. It proves counterfactual prediction benefits from simultaneously controlling the underlying confounders in observational features and unobserved confounders. The effectiveness of our disentanglement theory and hierarchical self-distillation framework is thus validated. The results in all Twins datasets are consistent with these findings.

**Continuous Scenario**: Following (Wu et al., 2022a), we use Demand-$\alpha$-$\beta$ to denote different data setting in Demands datasets. Demand-0-1 represents the original Demand dataset defined in (Hartford et al., 2017). The $\alpha$ in Demand-$\alpha$-$\beta$ denotes the extra information from instrumental variables while the $\beta$ indicates the information increasing from instrumental variables and underlying confounders together on the basis of Demand-0-1. We only present the results on Demand-0-1 with box plots as shown in Figure 3(a), where we omit the results of CEVAE as its $MSE$ is beyond 10000. **The detailed experimental results on all Demands datasets are provided in the Appendix A.2.2**.

In general, the IV-based methods perform worse than the non-IV-based ones under the continuous scenario. This result indicates that confounding bias from the treatment regression stage is a critical problem in IV-based methods, which also underscores the necessity of decomposing $C$ from observed variables $X$ and thereby correcting for confounding bias caused by $C$, coinciding with the findings under the binary scenario. We notice that DRCFR, which performs well on discrete datasets, suffers significantly on the Demand-0-1, reflecting the limitations of their disentanglement theory under the continuous scenario. Among all the baselines, $SD^2$ achieves the best and most stable performance on all Demands Datasets, demonstrating the generalizability of our algorithm on various types of datasets.

### 5.3.2 DISENTANGLEMENT VISUALIZATION

Similar to (Hassanpour & Greiner, 2019), we use the first(second) slice to denote the weight matrix that connects the variables in X belonging (not belonging) to the actual variables. The polygons' radii in Figure 3(b) quantify the average weights of the first slice (in red) and the second slice (in blue). We plot the radar charts to visualize the contribution of actual variables to the representations of each factor on $SD^2$ and another typical causal disentangled learning work DRCFR, a baseline that ensures the disentanglement of $A$ by proposing Prop. 3.1, yet does not provide disentangling solutions for $Z$ and $C$. As shown in each sub-figure in Figure 3(b), each vertex on the polygon represents the results of a synthetic dataset $m_v$-$m_z$-$m_c$-$m_a$-$m_u$. Compared with DRCFR, our method realizes much better identification performance of all three underlying factors in all datasets, indicating that our approach can achieve successful disentanglement performance.

Table 2: Performance comparison of bias of ATE between $SD^2$ and a method which replaces the disentanglement modules in $SD^2$ with one of the SOTA mutual information estimators $CLUB$.

| Method | 0-4-4-2-2 | 2-4-4-2-2 | 0-4-4-2-10 | 0-6-2-2-2 | Twins-0-16-8 | Twins-4-16-8 | Twins-0-16-12 | Twins-0-20-8 |
|--------|-----------|-----------|------------|-----------|--------------|--------------|---------------|--------------|
| **CLUB** | 0.517(0.004) | 0.513(0.006) | 0.563(0.004) | 0.487(0.005) | 0.016(0.013) | 0.026(0.009) | 0.021(0.012) | 0.018(0.010) |
| **Ours** | **0.012(0.008)** | **0.022(0.017)** | **0.029(0.018)** | **0.013(0.010)** | **0.012(0.007)** | **0.011(0.006)** | **0.012(0.006)** | **0.011(0.009)** |

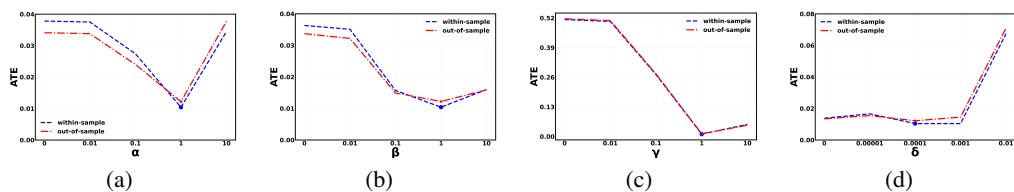

Figure 4: Hyper-parameters sensitivity analysis of $\alpha, \beta, \gamma, \delta$ on Syn-0-4-4-2-2 dataset. The blue and red lines show the ATE results of these parameters in within-sample and out-of-sample settings, respectively. The blue circle and red star represent the best parameters for the setting.

### 5.3.3 DIFFICULTY OF MUTUAL INFORMATION ESTIMATION

The difficulty of high-dimensional mutual information estimation has been widely discussed in (Poole et al., 2019; Belghazi et al., 2018; Cheng et al., 2020). To quantify this difficulty, we replace our core disentanglement module with one of the state-of-the-art mutual information estimators (CLUB (Cheng et al., 2020)) to directly minimize the mutual information between $R_z$, $R_c$, $R_a$. The related experimental results on out-of-sample cases ($CLUB$ in Table 2) show that the performance of our method gets impeded dramatically, indicating the importance of bypassing the complex mutual information estimator. We leave the results for within-sample cases in Appendix A.2.1.

### 5.3.4 HYPER-PARAMETERS ANALYSIS

With the multi-term total loss function shown in equation 10, we study the impact of each item on the counterfactual prediction on Syn-0-4-4-2-2 dataset. As can be seen from Figure 4(c), the performance of $SD^2$ is mostly affected by the changing in $\gamma$. In addition, although the performance fluctuates with the change of $\alpha$ and $\beta$, $SD^2$ performs better than almost all baselines. These two facts demonstrate that the improvement of inference performance is greatly contributed by the decomposition of $Z$ and $C$. Besides, from Figure 4(d), we find that the performance of the method is affected by changing $\delta$ as well, indicating the necessity of limiting the complexity of the model.

The ablation study, scalability analysis, used hardware and optimal hyper-parameters are provided in the Appendix A.2.3, A.2.4, A.3.3 and A.3.4.

## 6 CONCLUSION AND LIMITATIONS

To resolve the challenge of decomposing confounders and instrumental variables in causal disentangled representation learning, we provide a theoretically guaranteed solution to minimizing the mutual information, which fundamentally disentangles three types of underlying factors in observational features. On this basis, we design a hierarchical self-distilled disentanglement framework $SD^2$ to advance counterfactual prediction by eliminating the confounding bias caused by the observed and unobserved confounders simultaneously. Extensive experimental results on synthetic and real-world datasets validate the effectiveness of our proposed theory and framework in both binary and continuous data settings. **Limitations:** Due to the lack of real-world counterfactual datasets, $SD^2$ is only evaluated on limited tasks. It would be interesting to extend our method into other research areas, such as transfer learning (Rojas-Carulla et al., 2015), domain adaptation (Magliacane et al., 2017), counterfactual fairness (Kim et al., 2020), for robust feature extraction. We will leave future work for the generalization of our method to these fields.

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

## A  APPENDIX

### A.1  THEORETICAL PROOF

#### A.1.1  PROOF OF PROPOSITION 3.2

In the causal graph shown in Fig.1(a), instrumental variables $Z$ and confounders $C$ have the same dependence with treatment $T$, outcome $Y$ and adjustable variables $A$.

*Proof.* Based on the d-separation (Pearl et al., 2000), we consider the cases related to $T$, $Y$ and $A$, respectively.

1. **Cases related to $T$**

   Obviously, $Z$ and $C$ are direct parents of $T$. Therefore, they are both dependent on $T$ and conditional dependent on $T$ when any variables are conditioned on.

2. **Cases related to $Y$**

   Firstly, $C$ is a direct parent of $Y$, which means it is dependent on $Y$ and conditionally dependent on $Y$ conditioned on any variables.

   Secondly, the relations between $Z$ and $Y$ are as following:

   ① $Z \not\perp Y$ because of the chain structure $Z \to T \to Y$.

② $Z \not\perp Y \mid T$ as the collider structure $Z \to T \leftarrow C$ is opened when $T$ is conditioned on. The correlation flows from $Z$ to $C$ to $Y$.

③ $Z \not\perp Y \mid A$ similar with 2.①.

④ $Z \not\perp Y \mid A, T$ similar with 2.②.

3. **Cases related to $A$**

   The dependence between $Z, C$ with $A$ are as following:

   ① $Z \perp A$ because of the collider structure $T \to Y \leftarrow A$.

   $C \perp A$ because of the collider structure $C \to Y \leftarrow A$.

   ② $Z \perp\!\!\!\perp A \mid T$ and $C \perp\!\!\!\perp A \mid T$ because of the collider structure $C \to Y \leftarrow A$.

   ③ $Z \not\perp A \mid Y$ as the collider structure $T \to Y \leftarrow A$ is opened when $Y$ is conditioned on. The correlation flows from $Z$ to $T$ to $A$.

   $C \not\perp A \mid Y$ as the collider structure $C \to Y \leftarrow A$ is opened when $Y$ is conditioned on.

   ④ $Z \not\perp A \mid T, Y$ and $C \not\perp A \mid T, Y$ as the collider structures $T \to Y \leftarrow A$ and $C \to Y \leftarrow A$ are opened at the same time. $Z, C$ and $A$ become dependent on each other.

We have listed all the possible cases, showing that $Z$ and $C$ have the same dependence with $A$, $T$ and $Y$. Therefore, **Prop.3.2** holds. □

### A.1.2 PROOF OF THEOREM 4.1

Consider $R_a$ and $R_c$ as the representations of $A$ and $C$ produced by the representation networks, and let Y be the label of the outcome. We have,

$$\min I(Y; R_c) - I(R_c; Y \mid R_a) \iff \min H(Y) - H(Y \mid R_c) - H(Y \mid R_a) + H(Y \mid R_c, R_a).$$

*Proof.* Based on the definition of mutual information (Belghazi et al., 2018):

$$I(Y; R_c) = H(Y) - H(Y \mid R_c), \tag{12}$$

where $H(Y)$ denotes Shannon entropy, and $H(Y \mid R_c)$ is the conditional entropy of $Y$ given $R_c$. Similarly,

$$I(R_c; Y \mid R_a) = H(Y \mid R_a) - H(Y \mid R_a, R_c), \tag{13}$$

where $I(R_c; Y \mid R_a)$ denotes the conditional mutual information between $R_c$ and $Y$ given $R_a$; $H(Y \mid R_a)$ and $H(Y \mid R_a, R_c)$ is the the conditional entropy of $Y$ given $R_a$, $R_a$ and $R_c$, respectively.

Combining equation 12 and equation 13, we have **Theo.4.1** holds. □

### A.1.3 PROOF OF COROLLARY 4.2

One of sufficient conditions of minimizing $I(R_a; R_c)$ is:

$$\min \begin{cases} D_{KL}[\mathcal{P}_Y \| \mathcal{P}_Y^{R_a}] \\ D_{KL}[\mathcal{P}_Y^{R_a} \| \mathcal{P}_Y^{R_c}], \end{cases}$$

where $\mathcal{P}_Y^{R_a} = p(Y \mid R_a)$, $\mathcal{P}_Y^{R_c} = p(Y \mid R_c)$ represent the predicted distributions, $\mathcal{P}_Y = p(Y)$ represents the real distribution, and $D_{KL}$ denotes the KL-divergence.

*Proof.* Based on the definition of conditional entropy, for any continuous variables $R_c$, $R_a$ and $Y$, we have:

$$H(Y) - H(Y \mid R_c) - H(Y \mid R_a) + H(Y \mid R_c, R_a) =$$
$$- \int p(Y) log p(Y) dY$$
$$+ \underbrace{\int p(R_c) dR_c \int p(Y \mid R_c) log p(Y \mid R_c) dY}_{termC}$$
$$+ \underbrace{\int p(R_a) dR_a \int p(Y \mid R_a) log p(Y \mid R_a) dY}_{termA}$$
$$- \underbrace{\iint p(R_a, R_c) dR_a dR_c \int p(Y \mid R_a, R_c) log p(Y \mid R_a, R_c) dY}_{termM} .$$

$$(15)$$

We further inspect $termC$ in equation 15 and have:

$$\int p(R_c) dR_c \int p(Y \mid R_c) log p(Y \mid R_c) dY =$$
$$\iint p(R_c) p(Y \mid R_c) log \left[ \frac{p(Y \mid R_c)}{p(Y \mid R_a)} p(Y \mid R_a) \right] dR_c dY.$$

$$(16)$$

By factorizing the double integrals in equation 16 into another two components, we show the following:

$$\iint p(R_c) p(Y \mid R_c) log \left[ \frac{p(Y \mid R_c)}{p(Y \mid R_a)} p(Y \mid R_a) \right] dR_c dY =$$
$$\underbrace{\iint p(R_c) p(Y \mid R_c) log \frac{p(Y \mid R_c)}{p(Y \mid R_a)} dR_c dY}_{termC_1} + \underbrace{\iint p(R_c) p(Y \mid R_c) log p(Y \mid R_a) dR_c dY}_{termC_2} .$$

$$(17)$$

Conduct similar factorization for $termA$ and $termM$ in equation 15, we have:

$$\int p(R_a) dR_a \int p(Y \mid R_a) log p(Y \mid R_a) dY =$$
$$\underbrace{\iint p(R_a) p(Y \mid R_a) log \frac{p(Y \mid R_a)}{p(Y \mid R_c)} dR_a dY}_{termA_1} + \underbrace{\iint p(R_a) p(Y \mid R_a) log p(Y \mid R_c) dR_a dY}_{termA_2}$$

$$(18)$$

$$\iint p(R_a, R_c) dR_a dR_c \int p(Y \mid R_a, R_c) log p(Y \mid R_a, R_c) dY =$$
$$\underbrace{\iiint p(R_a, R_c) p(Y \mid R_a, R_c) log \frac{p(Y \mid R_a, R_c)}{p(Y \mid R_a)} dR_a dR_c dY}_{termM_1} +$$
$$\underbrace{\iiint p(R_a, R_c) p(Y \mid R_a, R_c) log p(Y \mid R_a) dR_a dR_c dY}_{termM_2} .$$

$$(19)$$

Integrate $termC_1$, $termA_1$ and $termM_1$ over $Y$:

$$C_1 = \int p(R_c) D_{KL} \left[ p(Y \mid R_c) \| p(Y \mid R_a) \right] dR_c, \tag{20}$$

$$A_1 = \int p(R_a) D_{KL} \left[ p(Y \mid R_a) \| p(Y \mid R_c) \right] dR_a, \tag{21}$$

$$M_1 = \iint p(R_a, R_c) D_{KL} \left[ p(Y \mid R_a, R_c) \| p(Y \mid R_a) \right] dR_a dR_c, \tag{22}$$

where $D_{KL}$ denotes KL-divergence. Integrate $termC_2$ and $termA_2$ over $R_c$ and $R_a$, respectively, we have:

$$C_2 = \int p(Y)logp(Y \mid R_a)dY, \tag{23}$$

$$A_2 = \int p(Y)logp(Y \mid R_c)dY. \tag{24}$$

Integrate $termM_2$ over $R_c$, we have:

$$M_2 = \iint p(Y, R_a)logp(Y \mid R_a)dR_adY. \tag{25}$$

We further factorize equation 25 into another two components:

$$
\begin{aligned}
& \iint p(Y, R_a)logp(Y \mid R_a)dR_adY \\
&= \iint p(Y, R_a)log\left[\frac{p(Y, R_a)}{p(R_a)}\right]dR_adY \\
&= \iint p(Y, R_a)logp(Y, R_a)dR_adY - \iint p(Y, R_a)logp(R_a)dR_adY \\
&= -H(Y, R_a) + H(R_a) \\
&= -H(Y \mid R_a).
\end{aligned}
\tag{26}
$$

In the view of above, we have the following:

$$
\begin{aligned}
& H(Y) - H(Y \mid R_c) - H(Y \mid R_a) + H(Y \mid R_c, R_a) = \\
& - \int p(Y)logp(Y)dY \\
& + \int p(R_c)D_{KL}\left[p(Y \mid R_c)\|p(Y \mid R_a)\right]dR_c + \int p(Y)logp(Y \mid R_a)dY \\
& - H(Y \mid R_a) \\
& - \iint p(R_a, R_c)D_{KL}\left[p(Y \mid R_a, R_c)\|p(Y \mid R_a)\right]dR_adR_c + H(Y \mid R_a).
\end{aligned}
\tag{27}
$$

Based on the non-negativity of KL-divergence, equation 27 is upper bounded by:

$$
\begin{aligned}
& - \int p(Y)logp(Y)dY + \int p(R_c)D_{KL}\left[p(Y \mid R_c)\|p(Y \mid R_a)\right]dR_c + \int p(Y)logp(Y \mid R_a)dY = \\
& \int p(R_c)D_{KL}\left[p(Y \mid R_c)\|p(Y \mid R_a)\right]dR_c + \int p(Y)log\left[\frac{p(Y \mid R_a)}{p(Y)}\right]dY.
\end{aligned}
\tag{28}
$$

Equivalently, we have the upper bound as:

$$\mathbb{E}_{R_a\sim E_\theta(R_a|X)}\mathbb{E}_{R_c\sim E_\phi(R_c|X)}[D_{KL}[p(Y \mid R_c)\|p(Y \mid R_a)]] + \mathbb{E}_{R_a\sim E_\theta(R_a|X)}\left[log\left[\frac{p(Y \mid R_a)}{p(Y)}\right]\right], \tag{29}$$

where $\theta, \phi$ denote the parameters of the representation networks of $A$ and $C$, respectively. Therefore, the objective of separating $C$ and $A$ from $X$ can be formalized as:

$$\min_{\theta,\phi}\mathbb{E}_{R_a\sim E_\theta(R_a|X)}\mathbb{E}_{R_c\sim E_\phi(R_c|X)}\left[D_{KL}[\mathcal{P}_Y^{R_a}\|\mathcal{P}_Y^{R_c}] + log\left[\frac{\mathcal{P}_Y^{R_a}}{\mathcal{P}_Y}\right]\right], \tag{30}$$

where $\mathcal{P}_Y^{R_a} = p(Y \mid R_a)$, $\mathcal{P}_Y^{R_c} = p(Y \mid R_c)$ and $\mathcal{P}_Y$ denote the predicted distributions of $Y$ from the representations $R_a$, $R_c$ and real distribution of $Y$, respectively.

Clearly, the first term in equation 30 is equivalent to minimize the discrepancy between the predicted distributions of $Y$ from the representations $R_a$, $R_c$. Notice the second term in equation 30 can be implicitly reduced by minimizing $D_{KL}\left[\mathcal{P}_Y^{R_a}\|\mathcal{P}_Y\right]$. Thus, we have:

$$\min \begin{cases} D_{KL}[\mathcal{P}_Y\|\mathcal{P}_Y^{R_a}] \\ D_{KL}[\mathcal{P}_Y^{R_a}\|\mathcal{P}_Y^{R_c}] \end{cases}$$

Table 3: Performance comparison of bias of ATE between $SD^2$ and a method which replaces the disentanglement modules in $SD^2$ with one of the SOTA mutual information estimators $CLUB$ for within-sample cases.

| Method | 0-4-4-2-2 | 2-4-4-2-2 | 0-4-4-2-10 | 0-6-2-2-2 | Twins-0-16-8 | Twins-4-16-8 | Twins-0-16-12 | Twins-0-20-8 |
|---|---|---|---|---|---|---|---|---|
| CLUB | 0.514(0.005) | 0.513(0.006) | 0.563(0.006) | 0.481(0.006) | 0.020(0.009) | 0.029(0.008) | 0.024(0.007) | 0.023(0.005) |
| Ours | **0.010(0.008)** | **0.017(0.013)** | **0.029(0.019)** | **0.014(0.013)** | **0.008(0.006)** | **0.007(0.004)** | **0.007(0.006)** | **0.008(0.006)** |

Table 4: Performance comparison of bias of ATE between $SD^2$ and the SOTA baselines on the synthetic datasets ($mv$-$mz$-$mc$-$ma$-$mu$) and Twins datasets (Twins-$mv$-$mx$-$mu$) in within-sample setting. Bold/Underline indicates the method with the best/second-best performance.

| Within-Sample | Synthetic | | | | Twins | | | |
|---|---|---|---|---|---|---|---|---|
| Method | 0-4-4-2-2 | 2-4-4-2-2 | 0-4-4-2-10 | 0-6-2-2-2 | Twins-0-16-8 | Twins-4-16-8 | Twins-0-16-12 | Twins-0-20-8 |
| DirectRep | 0.037(0.023) | 0.034(0.009) | 0.060(0.014) | 0.043(0.025) | 0.015(0.007) | 0.009(0.008) | 0.018(0.012) | 0.012(0.007) |
| CFR | 0.031(0.014) | 0.032(0.017) | 0.056(0.017) | 0.060(0.019) | 0.011(0.010) | 0.012(0.008) | 0.017(0.014) | 0.010(0.011) |
| DFL | 3.696(0.034) | 3.702(0.045) | 4.055(0.041) | 3.450(0.030) | 0.173(0.018) | 0.181(0.014) | 0.169(0.020) | 0.175(0.019) |
| DRCFR | 0.058(0.018) | 0.050(0.025) | 0.071(0.020) | 0.066(0.015) | 0.015(0.016) | 0.008(0.005) | 0.016(0.017) | 0.024(0.017) |
| DEVAE | 0.424(0.009) | 0.432(0.007) | 0.373(0.004) | 0.461(0.008) | 0.037(0.010) | 0.034(0.006) | 0.030(0.009) | 0.026(0.007) |
| DeepIV-Log | 0.569(0.024) | 0.567(0.011) | 0.601(0.011) | 0.531(0.019) | 0.019(0.010) | 0.017(0.016) | 0.022(0.011) | 0.012(0.007) |
| DeepIV-Gmm | 0.466(0.012) | 0.387(0.021) | 0.518(0.012) | 0.437(0.007) | 0.017(0.011) | 0.013(0.009) | 0.017(0.011) | 0.013(0.011) |
| DFIV | 3.947(0.108) | 3.810(0.041) | 4.400(0.134) | 3.644(0.103) | 0.257(0.145) | 0.255(0.148) | 0.256(0.144) | 0.227(0.151) |
| OneSIV | 0.504(0.008) | 0.441(0.063) | 0.569(0.013) | 0.478(0.012) | 0.016(0.014) | 0.011(0.007) | 0.025(0.020) | 0.013(0.010) |
| CBIV | 0.063(0.025) | 0.065(0.032) | 0.049(0.022) | 0.033(0.023) | 0.024(0.016) | 0.057(0.012) | 0.051(0.038) | 0.037(0.019) |
| Ours | **0.010(0.008)** | **0.017(0.013)** | **0.029(0.019)** | **0.014(0.013)** | **0.008(0.006)** | **0.007(0.004)** | **0.007(0.006)** | **0.008(0.006)** |

$$\Rightarrow \min H(Y) - H(Y \mid R_c) - H(Y \mid R_a) + H(Y \mid R_c, R_a).$$

**Corol.4.2** holds. □

## A.2 Additional Experiments

### A.2.1 Results of Within Samples

We present the experimental results of within-sample cases for binary settings in Table 4 and for Section 5.3.3 in Table 3, which are consistent with the findings in the main paper.

### A.2.2 Experiments on Continuous Setting

The experimental results on Demands datasets are presented in Table 5. It can be seen that whether adding the information of instrumental variables (Demands-5-1) or increasing the information of confounders (Demands-0-5), our algorithm performs far better than all the baseline, which proves the efficiency and generalizability of our method.

### A.2.3 Ablation Study

We perform the ablation experiments to examine the contributions of each component in total loss function on final inference performance. We test the performance improvement of treatment regression , adjustable variable decomposition, instrument variable and confounder decomposition to the final effect. The results are shown in Table 6.

Firstly, we only preserve representation networks and deep outcome classifier in Figure 2. The objective loss is reduced to factual loss $L_p$ plus regularization loss. Secondly, we add the deep treatment classifier into the model on the basis of the first model, while the objective loss becomes $L_p + L_t$. Then, the adjustable variables decomposition module is integrated into the second model, thus the objective loss is $L_p + L_t + L_a$. To demonstrate the difficulty of mutual information es-

Table 5: Performance comparison of MSE between $SD^2$ and the SOTA baselines on the Demands datasets (Demands-$\alpha$-$\beta$). Bold/Underline indicates the method with the best/second-best performance.

| Method | Within-Sample | | | Out-of-Sample | | |
|---|---|---|---|---|---|---|
| | Demands-0-1 | Demands-0-5 | Demands-5-1 | Demands-0-1 | Demands-0-5 | Demands-5-1 |
| **DirectRep** | 472.14(292.68) | 319.27(156.58) | 795.38(304.10) | 494.47(287.92) | 303.22(144.85) | 768.15(278.24) |
| **CFR** | 1180.26(439.56) | 340.12(175.82) | 1891.91(79.78) | 1103.00(362.45) | 326.54(110.31) | 1877.60(119.71) |
| **DFL** | 786.92(122.35) | 1074.71(142.78) | 874.78(103.71) | 736.35(123.44) | 914.08(90.11) | 822.71(65.88) |
| **DRCFR** | 720.51(195.09) | 451.31(407.55) | 809.04(94.98) | 766.22(188.86) | 498.22(363.25) | 864.21(83.39) |
| **DEVAE** | >10000 | >10000 | >10000 | >10000 | >10000 | >10000 |
| **DeepIV-Gmm** | 1774.71(757.98) | 3429.79(438.65) | 2618.72(775.20) | 904.60(618.41) | 2423.33(326.23) | 2405.33(675.40) |
| **DFIV** | 862.00(123.71) | 1420.24(257.10) | 909.25(137.05) | 943.57(188.70) | 1213.27(388.08) | 1050.65(253.38) |
| **OneSIV** | 2573.15(157.31) | >10000 | 2352.69(125.16) | 2744.87(182.35) | 4243.51(596.10) | 2538.77(147.29) |
| **CBIV** | 819.52(535.05) | 410.42(241.71) | 2542.58(442.26) | 2990.19(735.92) | 316.66(114.89) | 2958.70(614.26) |
| **Ours** | **170.55(6.51)** | **216.55(47.11)** | **171.07(6.41)** | **183.21(14.18)** | **199.44(14.60)** | **187.20(16.05)** |

| Datasets | Within-Sample | | | | |
|---|---|---|---|---|---|
| | $L_p$ | $L_p + L_t$ | $L_p + L_t + L_a$ | $CLUB$ | $Total$ |
| **Twins-0-16-8** | 0.027(0.012) | 0.025(0.007) | 0.021(0.007) | 0.020(0.009) | **0.008(0.006)** |
| **Twins-4-16-8** | 0.067(0.078) | 0.022(0.005) | 0.024(0.005) | 0.029(0.008) | **0.008(0.005)** |
| **Twins-0-16-12** | 0.113(0.110) | 0.027(0.005) | 0.024(0.005) | 0.024(0.007) | **0.007(0.006)** |
| **Twins-0-20-8** | 0.031(0.010) | 0.020(0.004) | 0.023(0.006) | 0.023(0.005) | **0.008(0.006)** |
| **Syn-0-4-4-2-2** | 0.519(0.005) | 0.513(0.005) | 0.513(0.004) | 0.514(0.005) | **0.010(0.008)** |
| **Syn-2-4-4-2-2** | 0.517(0.007) | 0.513(0.007) | 0.513(0.007) | 0.513(0.006) | **0.017(0.013)** |
| **Syn-0-4-4-2-10** | 0.569(0.005) | 0.565(0.004) | 0.565(0.005) | 0.563(0.006) | **0.029(0.019)** |
| **Syn-0-6-2-2-2** | 0.486(0.004) | 0.483(0.004) | 0.483(0.004) | 0.481(0.006) | **0.015(0.013)** |

| Datasets | Out-of-Sample | | | | |
|---|---|---|---|---|---|
| | $L_p$ | $L_p + L_t$ | $L_p + L_t + L_a$ | $CLUB$ | $Total$ |
| **Twins-0-16-8** | 0.022(0.011) | 0.020(0.011) | 0.018(0.010) | 0.016(0.013) | **0.012(0.007)** |
| **Twins-4-16-8** | 0.059(0.076) | 0.017(0.012) | 0.019(0.012) | 0.026(0.009) | **0.009(0.006)** |
| **Twins-0-16-12** | 0.111(0.109) | 0.022(0.011) | 0.019(0.012) | 0.021(0.012) | **0.012(0.006)** |
| **Twins-0-20-8** | 0.026(0.013) | 0.015(0.009) | 0.018(0.012) | 0.018(0.010) | **0.011(0.009)** |
| **Syn-0-4-4-2-2** | 0.522(0.004) | 0.517(0.004) | 0.516(0.004) | 0.517(0.004) | **0.012(0.008)** |
| **Syn-2-4-4-2-2** | 0.517(0.006) | 0.512(0.007) | 0.513(0.006) | 0.513(0.006) | **0.022(0.017)** |
| **Syn-0-4-4-2-10** | 0.568(0.004) | 0.565(0.004) | 0.564(0.004) | 0.563(0.004) | **0.029(0.018)** |
| **Syn-0-6-2-2-2** | 0.492(0.005) | 0.489(0.005) | 0.489(0.005) | 0.487(0.005) | **0.013(0.010)** |

Table 6: Ablation Study. $L_p$ represents preserving representation networks and deep outcome classifier only. $L_p + L_t$ adds deep treatment classifier on the basis of the model of $L_p$. $L_p + L_t + L_a$ adds the adjustable variable decomposition module on the basis of the model of $L_p + L_t$. $CLUB$ adds one mutual information estimator on the basis of $L_p + L_t + L_a$. $Total$ model is our proposed model.

timation, we add one of the state-of-the art mutual information estimator (Cheng et al., 2020) into the third model and denote this model as $CLUB$. Finally, we introduce our shallow treatment and outcome classifiers into the third model, the objective loss of which is presented in equation 10 in the main paper and named as $Total$.

**Main Results**: We have the following observations from the experimental results in Table 6: (1) The model with loss function $L_p$ performs the worst for all data settings on Twins and Synthetic

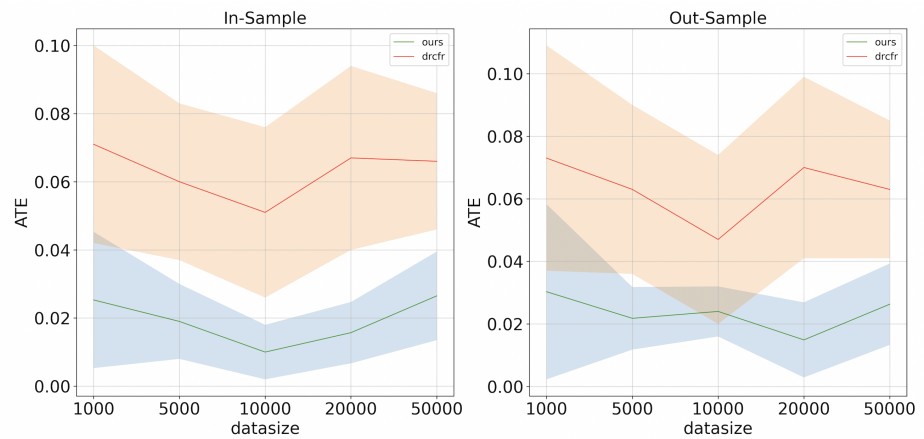

Figure 5: Scalability Analysis of $SD^2$.

datasets, demonstrating the significance of treatment prediction for counterfactual prediction in the presence of unobserved confounders. (2) If we only decompose adjustable variables $A$ from observed features $X$, as the results under the loss $L_p + L_t + L_a$ shows, the model indeed does not get much improvement on the inference performance compared with the results under the loss $L_p + L_t$. This may be due to the fact that only decomposing A cannot eliminate the confounding bias caused by the unmeasured confounders $U$ and underlying confounders existing in observed features $X$. In addition, it indicates the necessity of the decomposition of $C$ and $Z$ for counterfactual prediction. (3) When added shallow classifiers of treatment and outcome during training, as the results of $Total$ model show, our framework $SD^2$ achieves the best performance with the improvement of more than $60\%$ on Synthetic datasets and over $95\%$ on Twins datasets on the basis of the model under the loss $L_p + L_t + L_a$, which demonstrates the decomposition of $C$ and $Z$ indeed advance the counterfactual inference.

### A.2.4 SCALABILITY ANALYSIS

To demonstrate the scalability of our algorithm, we conduct experiments on Syn-0-4-4-2-2 with different sample sizes with our method and one excellent causal disentangled learning method DRCFR (Hassanpour & Greiner, 2019). Figure 5 shows the results of the related experimental results.It can be seen from the results that the performance of $SD^2$ remains stable and efficient with the variation of sample size in the dataset, which demonstrates the scalability of our algorithm.

### A.3 REPRODUCIBILITY

#### A.3.1 DATA GENERATION

We generate the synthetic datasets according to the following steps:

- For $K$ in $Z$, $C$, $A$, $V$, $U$, sample $K$ from $\mathcal{N}(0, I_{mk})$, where $I_{mk}$ denotes $mk$ degree identity matrix. Concatenate $Z$, $C$, and $A$ to constitute the observed covariates matrix $X$. $V$ represents the observed IVs, the dimension of which could be set to 0. The setting of $V$ is mainly for the comparison of IV-based methods. $U$ represents the unobserved confounders.

- Sample treatment variables $T$ as following:

$$T_i \sim Bern(Sigmoid(\\ \sum_{i=1}^{mz} Z_i + \sum_{i=1}^{mc} C_i + \sum_{i=1}^{mv} V_i + \sum_{i=1}^{mu} U_i)), \tag{32}$$

where $mz, mc, mv, mu$ represent the dimension of $Z, C, V, U$ respectively.

- Sample outcome variables $Y$ as following:

$$Y_i \sim Bern(Sigmoid(($$

$$\frac{Ti}{ma + mc + mu}\sum_{i=1}^{ma}{A_i}^2 + \sum_{i=1}^{mc}{C_i}^2 + \sum_{i=1}^{mu}{U_i}^2)+ \tag{33}$$

$$(\frac{1 - Ti}{ma + mc + mu}\sum_{i=1}^{ma}A_i + \sum_{i=1}^{mc}C_i + \sum_{i=1}^{mu}U_i))),$$

where $ma$ represents the dimension of $A$.

### A.3.2  LOSS FUNCTIONS FOR CONTINUOUS SCENARIO

The loss function for disentangling A is defined as following:

$$\mathcal{L}_a = \beta_1 L(\hat{T}_c, T) + \beta_2 KL(\hat{T}_c, \hat{T}) + \beta_3 KL(\hat{T}_c, \hat{T}_a), \tag{34}$$

where $\hat{T}_{variable}$ in $L(\cdot)$ represents the predicted values for $variable$ while in $KL(\cdot)$ it denotes the distribution of the variable. Same below.

We assume that the continuous variable follows a normal distribution, therefore the KL divergence can be calculated with:

$$\text{KL}(q\|p) = \log \sigma_2 - \log \sigma_1 + \frac{\sigma_1^2 + (\mu_1 - \mu_2)^2}{2\sigma_2^2} - \frac{1}{2}, \tag{35}$$

where $\mathbf{q} \sim \mathcal{N}\left(\mu_1, \sigma_1^2\right), \mathbf{p} \sim \mathcal{N}\left(\mu_2, \sigma_2^2\right)$.

The loss function for disentangling $Z$ and $C$ are shown in equation 36 and equation 37:

$$\mathcal{L}_z = L(Q_Y^{z;t}, Y) + L(Q_Y^t, Y) + L(Q_Y^{z;t}, Q_Y) + L(Q_Y^t, Q_Y) + L(Q_Y^{z;t}, Q_Y^t), \tag{36}$$

$$\mathcal{L}_c = L(Q_T^z, T) + L(Q_T^c, T) + L(Q_T^c, Q_T) + L(Q_T^z, Q_T) + L(Q_T^c, Q_T^z) \\ + L(Q_Y^c, Y) + L(Q_Y^a, Y) + L(Q_Y^c, Q_Y) + L(Q_Y^a, Q_Y) + L(Q_Y^c, Q_Y^a). \tag{37}$$

To reduce the confounding bias led by observed confounders, we define the following loss function:

$$\mathcal{L}_{oc} = (\hat{T}_z, T) + (\hat{T}_z, \hat{T}) + (\hat{T}_z, \hat{T}e), \tag{38}$$

where $\tilde{c}$ denotes the representation of $C$ after re-balance network.

Therefore, the total loss function of $SD^2$ for the continuous scenario can be devised as:

$$\mathcal{L}_{SD^2} = \underbrace{L(\hat{Y}, Y) + \alpha L(\hat{T}, T)}_{factual\ loss} + \underbrace{\beta \mathcal{L}_a + \gamma(\mathcal{L}_c + \mathcal{L}_z)}_{disentanglement\ loss} + \underbrace{\omega \mathcal{L}_{oc}}_{rebalance\ loss} + \underbrace{\delta \|W\|_2}_{regularization\ loss}. \tag{39}$$

### A.3.3  OPTIMAL HYPER-PARAMETERS

Optimal hyper-parameters are presented in Table 7 and Table 8.

### A.3.4  HARDWARE

In this work, we perform all experiments on a cluster with two 12-core Intel Xeon E5-2697 v2 CPUs and a total 768 GiB Memory RAM.

| Hyper-parameters | Synthetic | | | | Twins | | | |
|---|---|---|---|---|---|---|---|---|
| | **0-4-4-2-2** | **2-4-4-2-2** | **0-4-4-2-10** | **0-6-2-2-2** | **0-16-8** | **4-16-8** | **0-16-12** | **0-20-8** |
| Learning rate | 5e-4 | 5e-4 | 5e-4 | 5e-4 | 5e-4 | 5e-4 | 5e-4 | 5e-4 |
| $\alpha$ | 1 | 1 | 1 | 1 | 0.1 | 0.1 | 0.1 | 1 |
| $\beta$ | 1 | 1 | 1 | 1 | 0.01 | 0.01 | 0.01 | 0.001 |
| $\gamma$ | 1 | 1 | 1 | 1 | 0.1 | 0.1 | 0.1 | 1 |
| $\delta$ | 1e-4 | 1e-4 | 1e-4 | 1e-4 | 1e-4 | 1e-4 | 1e-4 | 1e-4 |
| Temperature | 1 | 1 | 1 | 1 | 1 | 1 | 1 | 1 |
| Depth of representation network | 3 | 3 | 3 | 3 | 3 | 3 | 3 | 3 |
| Depth of treatment classifier | 2 | 2 | 2 | 2 | 2 | 2 | 2 | 2 |
| Depth of outcome classifier | 3 | 3 | 3 | 3 | 3 | 3 | 3 | 3 |
| Dim of representation network | $(256)_3$ | $(256)_3$ | $(256)_3$ | $(256)_3$ | $(256)_3$ | $(256)_3$ | $(256)_3$ | $(256)_3$ |
| Dim of treatment classifier | 128-64 | 128-64 | 128-64 | 128-64 | 128-64 | 128-64 | 128-64 | 128-64 |
| Dim of outcome classifier | $(128)_3$ | $(128)_3$ | $(128)_3$ | $(128)_3$ | $(128)_3$ | $(128)_3$ | $(128)_3$ | $(128)_3$ |
| Iteration | 2000 | 2000 | 2000 | 2000 | 1000 | 1000 | 2000 | 300 |
| Batch size | 100 | 100 | 100 | 100 | 100 | 100 | 100 | 100 |

Table 7: Optimal hyper-parameters for binary datasets.

| Hyper-parameters | **Demands-0-1** | **Demands-0-5** | **Demands-5-1** |
|---|---|---|---|
| Learning rate | 1e-3 | 1e-3 | 1e-3 |
| $\alpha$ | 0.01 | 0.1 | 0.01 |
| $\beta$ | 0.1 | 0.1 | 0.1 |
| $\gamma$ | 1 | 0.1 | 1 |
| $\omega$ | 0.1 | 0.1 | 0.1 |
| $\delta$ | 1e-4 | 1e-4 | 1e-4 |
| Depth of representation network | 3 | 3 | 3 |
| Depth of treatment regressor | 3 | 3 | 3 |
| Depth of outcome regressor | 3 | 3 | 3 |
| Dim of representation network | $(256)_3$ | $(256)_3$ | $(256)_3$ |
| Dim of treatment classifier | $(64)_3$ | $(64)_3$ | $(64)_3$ |
| Dim of outcome classifier | $(64)_3$ | $(64)_3$ | $(64)_3$ |
| Iteration | 10000 | 10000 | 10000 |
| Batch size | 100 | 100 | 100 |

Table 8: Optimal hyper-parameters for continuous datasets.

