# OpenReview forum: "Self-Distilled Disentanglement for Counterfactual Prediction"
_ICLR.cc/2024/Conference — Submitted to ICLR 2024_

### Official Review · Reviewer_YeKP · 2023-10-28

**Soundness:** 3 good
**Presentation:** 3 good
**Contribution:** 3 good
**Rating:** 6
**Confidence:** 3

**Summary:**

This paper proposes two novel theorems (Theorems 4.1 & 4.2) to disentangle the representations of instrumental variables, confounders, and adjustable variables from pre-treatment variables and bypass MI estimation between high-dimensional representations from the perspective of information theory.

**Strengths:**

- This paper considers an important problem in causal inference and proposes two novel theorems to disentangle the representations of instrumental variables, confounders, and adjustable variables from pre-treatment variables. The results demonstrate the effectiveness of the proposed algorithm.

- This paper provides a comprehensive review of the literature on counterfactual prediction and disentangled representation learning. The paper is well-organized.

**Weaknesses:**

Incorrect Definiton about $I(A; B \mid C)$, which should not denote conditional mutual information. If $I(Z ; Y\mid T)$ represents conditional mutual information, then in Eqs. (2,3,4), it is evident that the conditional mutual information $I(Z ; Y \mid T) ≠ 0$, as the open of the collider structure $Z → T ← \\{C, U\\}$ will make $Z$ dependent on $\\{C, U\\}$ when $T$ is fixed as a condition, which consequently results in the dependence of Z and $Y$ through $\\{C, U\\}$. Then the authors use the conditional mutual information in the chain rule of mutual information again. The authors should clarify this point and differentiate the definition $I(Z ; Y \mid T)$ in Eqs. (2,3,4) from the definition of conditional mutual information. The relevant content may need to be restated.

**Questions:**

- Is it necessary to use a shallow network for $Q^z_T$ and $Q^c_T$? Why not use a network of the same size as the reference network?

- The optimization directions of the losses $L\left(Q_T^z, T\right) + L\left(Q_T^c, T\right)$ and $L\left(Q_T^c, Q_T^z\right)$ may be different or conflicting because the former aims to maximize the predictive abilities of $z$ and $c$ on $T$ (the predictive abilities of $c$ and $z$ on $T$ are different), while the latter actually implies forcing a better-performing model to reduce its predictive abilities. This can be achieved by modifying either the predictive network or the representation network. In essence, it is a non-zero-sum game problem. Only when all three are equal to 0 does it mean that $D[\mathcal{P}_T^{R_z} \| \mathcal{P}_T^{R_c}]=0$.
Otherwise, minimizing the loss of self-distilled disentanglement does not necessarily mean minimizing $D[\mathcal{P}_T^{R_z} \| \mathcal{P}_T^{R_c}]$. I am not sure if I have missed any important parts, but it clearly requires further clarification. Additionally, the Teacher network only aims at enhancing information prediction abilities and does not seem to be the focus of this paper? I will adjust my scores based on the author's response.

---

> ### Author Response · Authors · 2023-11-20
> **Response to Weaknesses and Questions**
>
> **Weaknesses - Response:**
>
> Firstly, we acknowledge there is an issue with the interpretation of the disentanglement process for variable Z within the context of the associated loss function in Formula 4. Our initial concept was grounded in the understanding that when using the treatment label T as the feature to predict outcome Y, the information in Z might become redundant. Therefore, we sought to disentangle Z from X by minimizing the KL-divergence between the predicted distribution of Y conditioned on both Rz and T, and the predicted distribution of Y conditioned solely on T, i.e., ${D}_{KL}\left[ p(Y \mid {Z},{T}) \Vert p(Y \mid T) \right]$. This served as the motivation for the design of the loss function used in subsequent experiments, and empirical studies have demonstrated its effectiveness. However, we recognize the inadequacy of the previous explanation. Therefore, in the updated version, we have removed the misleading description of the conditional mutual information $I (Z; Y|T)$. We appreciate your insightful feedback!
>
> **Questions - Response:**
>
> - **The Necessity of the Shallow Network:** A shallow network design for Z and C is necessary. As shown in Equation 11, we aim to directly obtain the predicted distribution of T based on Z and C, facilitating the disentanglement of C. This predicted distribution is precisely achieved by designing two shallow networks for Z and C. The retain network consolidates shared information from Z and C, and therefore, it should incorporate more parameters. A deep network is subsequently introduced to output predictions. This is the rationale behind our design of different shallow and deep networks.
>
> - **Clarification on the Loss Function:** We didn't specify whether it's a zero-sum or non-zero-sum problem. In theory, to minimize the mutual information between $R_z$ and $R_c$, we theoretically only need to minimize either $D_{KL}[P^{R_z}\_{T}|P_T]$ or $D_{KL}[P^{R_c}\_{T}| P_T]$ (we elaborate on choosing $D_{KL}[P^{R_z}\_{T}|P_T]$ for clarity) and $D_{KL}[P^{R_z}\_{T}|P ^{R_c}\_T]$. The corresponding loss functions are $L (Q^Z\_T, T)$ and $L (Q^Z\_T, Q^C\_T)$. However, in practice, including only these two components in the model loss makes convergence challenging, resulting in unsatisfactory performance. We speculate that this is due to a lack of sufficient supervised information guiding the updating direction of the prediction network for $R_c$. Therefore, we introduce the minimization of $D_{KL}[P^{R_c}\_{T}| P_T]$ (corresponding loss function $L (Q^C_T, T)$) to expedite loss convergence. Empirical results indicate that this loss function design improves disentanglement and counterfactual prediction performance, suggesting that, while not precise, it helps the model approach the optimum more quickly. We acknowledge that the original theoretical statements were not rigorous and have modified formulas 10 and 11 in section 4.2, further explaining loss functions in Section 4.3 in the updated version.
>
>     Your understanding of the teacher network is correct. We can set up a deep prediction network and and use its supervision information to train the shallow prediction models. From this perspective, our approach is essentially a self-distillation model and that’s why we name our method as **S**elf-**D**istilled **D**isentanglement, i.e., $SD^2$.  Therefore, the teacher model is utilized to enhance the network's prediction capability, thereby facilitating the distillation and disentanglement process.

---

> > ### Comment · Reviewer_YeKP · 2023-11-21
> >
> > I appreciate the author's response, and I intend to maintain my score as is. The score for this paper is marginally above the acceptance threshold. The paper introduces two interesting theorems (Theorems 4.1 and 4.2) aimed at identifying the representations of instrumental variables, confounders, and adjustable variables from pre-treatment variables.
> >
> > This work represents a commendable attempt in the field of causal disentanglement. However, the transition from theory to methodology in this paper seems not very smooth. Following the author's response, I agree that using self-distillation networks might provide a more promising estimation framework. But, the neural framework's depth in this paper, consisting of only 2 or 3 layers in prediction networks (Tables 7 & 8), appears limited. In such instances, would employing a shallower network as students to retain the model be beneficial?
> >
> > Given that the neural framework used in the experimental section of this paper is not originally deep (with only 2/3 layers), is it necessary to employ an even shallower network in this scenario?

---

> > > ### Author Response · Authors · 2023-11-22
> > >
> > > We intentionally opted for a relatively shallow neural framework with 2 or 3 layers in each network (deep, retain, shallow). This choice prioritized computational efficiency and addressed concerns related to overfitting, which is crucial for tasks involving large datasets or limited computational resources.
> > >
> > > Our distillation network is designed to improve the predictive capabilities of the shallow network. Let's delve into the distillation framework aimed at minimizing the loss $L_c^z$​. The teacher network consists of both the retain network and the deep network, utilizing Z and C as joint input features. In contrast, the student network, known as the shallow network, is only half the size of the teacher network and takes either Z or C as its input feature. Although the teacher network may not be large in terms of absolute parameter count, its prediction heads contain a greater amount of predictive information. This information is substantial enough to guide the training of the student network, thereby enhancing its predictive capabilities.

---

### Official Review · Reviewer_wJ78 · 2023-10-30

**Soundness:** 2 fair
**Presentation:** 4 excellent
**Contribution:** 3 good
**Rating:** 6
**Confidence:** 4

**Summary:**

This paper addresses the task of disentangling the underlying factors of observational datasets in causal inference. The paper proposes a novel disentanglement method that is capable of dissecting instrumental variables and confounders. The authors provide theoretical guarantees for their solution. They also evaluate their proposed method by conducting extensive experiments and show empirically that it outperforms SOTA.

**Strengths:**

- The ideas in the paper are presented clearly; it’s an easy paper to read and follow.
- The paper provides a good coverage of the related literature and clearly points out its contribution to the research area.
- Great use of probabilistic graphical models to clearly motivate the proposed solution.
- The idea of using mutual information to address disentanglement, as well as handling its challenges is interesting.
- The experiments are extensive and cover a wide range of scenarios.

**Weaknesses:**

- Some captions are not descriptive enough of the contents of the figures. E.g., Fig. 1(b) and (c).
- Use of inline equations should be avoided if possible.
- It’s best to state each finding in a separate bullet-point.
- When referring to the appendix, it’s best to also indicate its section number, to make it easier to find.

**Questions:**

- I’m not quite sure about the method’s name, specifically, what is being “distilled” here? Is this referring to dissected factors from X?
- What is being measured in the radar charts in Figure 3(b)? The caption states it shows “the contribution of actual and other variables to the decomposed representations”; how is this measured?
- It is stated in the paper that “the IV-based methods perform worse than the non-IV-based ones under the continuous scenario” but no reference or discussion is included. Please elaborate why.

---

> ### Author Response · Authors · 2023-11-20
> **Response to Weaknesses and Questions**
>
> **Weaknesses - Response:**
>
> Thank you for your advice. We have incorporated your suggestions into the updated version of the paper by adding more captions for Figure 1, transcribing all inline equations into display equations, breaking down the findings into separate bullet points and stating specifying the section numbers in the appendix when referenced, aiming to improve the paper's readability.
>
> **Questions - Response:**
>
> - **Method's name:** Here, "distillation" refers to a framework design that utilizes deep classifiers (for binary scenarios) or regressors (for continuous scenarios) to guide shallow ones in knowledge extraction, thereby promoting disentanglement performance.
>
>     Technically, the optimization of equation (8a, 8b, 9a, 9b) can be realized by training prediction networks about $Y$. Additionally, we can set up a deep prediction network from $T$, $A$, $C$ to $Y$ and use its supervision information to train the shallow prediction models. From this perspective, our approach is essentially a self-distillation model. Therefore, we name our method as **S**elf-**D**istilled **D**isentanglement, i.e., $SD^2$.
>
> - **Metrics in radar charts:** Following [2] and [3], we use the first (second) slice to denote the weight matrix that connects the variables in X belonging (not belonging) to the actual variables. The polygons’ radii in Figure 3b are scaled between 0: 0.09 and quantify the average weights of the first slice (in red) and the second slice (in blue). For better readability, we have added the description for this metric in Section 5.3.2. Thanks for the question.
>
> - **Discussion under the continuous scenario:** This outcome highlights the significant issue of confounding bias stemming from the treatment regression stage in IV-based methods. It emphasizes the essential need to disentangle confounding factors ($C$) from observed variables ($X$) and subsequently address confounding bias led by $C$. We appreciate your question, and we have integrated this discussion into the updated version.
>
> [2]. Negar Hassanpour and Russell Greiner. Learning disentangled representations for counterfactual regression. In International Conference on Learning Representations, 2019.
>
> [3]. Learning decomposed representation for counterfactual inference. arXiv preprint arXiv:2006.07040, 2020.

---

> ### Comment · Reviewer_wJ78 · 2023-12-02
> **assumptions should be tight enough**
>
> I thank the authors for their responses.
>
> After carefully reading the other reviews and the authors’ responses, I think the point raised by reviewer FUp5 is very important and must be properly addressed in the paper. It is necessary that their assumptions are tight enough that renders that counterexample not applicable.
>
> In light of this, I have decided to reduce my score.

---

### Official Review · Reviewer_pgv6 · 2023-10-31

**Soundness:** 3 good
**Presentation:** 2 fair
**Contribution:** 3 good
**Rating:** 3
**Confidence:** 3

**Summary:**

This paper proposes a method for representation learning for causal inference building on an existing disentanglement-based approach, by giving a method for optimizing the mutual information between two components of the representation without engaging in an intractible high-dimensional mutual information estimation problem. The authors discuss how to do this optimization by MI estimation with distillation problems and prove equivalencies in optimization problems. Empirically, they demonstrate that their method is better able to estimate ATEs, and through ablations show that their disentanglement is successful and that there is an advantage to avoiding the MI estimation step with their method.

**Strengths:**

- nice empirical results, showing a good win for their more flexible method as well as disentanglement results in controlled settings
- seems like some clever methods for avoiding problematic MI estimation
- draws good connections between deep learning approaches and causal formulations

**Weaknesses:**

My main issue with this paper is around clarity - I don't quite follow the novel/interesting parts, specifically through Sec 4. I think a lot of extra care could be taken rewriting here and could result in a nice paper. Specifically:
- Eq 3: both parts confuse me. On the left: it seems like the constraint I(Z; Y | T) = 0 contradicts the statement from the bottom of p3 about this inducing dependence through a collider. on the right: I'm not sure where this inequality constraint comes from, or why it's necessary
- Eq 4: this notation is confusing to me: I'm not clear on what variable is being minimized here. I think this could benefit from extra clarity spelling this out. Additionally, isn't I(Z, Y | T) a constant? should this be R_z? what precisely is the difference?
- What does it precisely mean to say that "the mutual information between R_a and R_c is all related to Y during the training phase"? And how would this be ensured by setting up prediction models that go out of R?
- Corollary 4.3: I don't understand this notation (10a-c, 11a-c)- does this mean minimizing all 3? minimizing the smallest? Again, I think clearer notation could be used around statements around optimization
- Sec 4.2: I don't follow a lot of these architectural choices: I think the "retain network" and "teacher networks" could have their roles explained more, as well as the relationship between the deep and shallow networks.
- Eq 12: a lot of this notation seems a little messy and leaves me uncertain: is W defined anywhere? the sampling weights w, and hyperparameters \alpha and \beta all look like global scalars in L_SD2, but I know that w should be w_i (example-wise weights), and so I'm not sure about \alpha and \beta - again these aren't defined anywhere
- what is the metric in 3b representing "contribution of actual variables?"


Smaller notes:
- top of p5: you say A is independent of C and Z - do you mean here that C is independent of A and Z? that seems to align more closely with the topic of the paragraph.
- I find the notation mk, mz, etc. to be confusing - these look like products to me rather than single variables. Maybe consider using m_k etc.
- it would be good to see more information on the contrast between your method and DRCFR

**Questions:**

- is there an implicit assumption in this method that T, Y are causally downstream of X? what other assumptions are required for this method to work well?

---

> ### Author Response · Authors · 2023-11-20
> **Response to Weaknesses and Questions**
>
> **Weaknesses-Response:**
>
> - Eq 3: Firstly, we acknowledge there is an issue with the interpretation of the disentanglement process for variable Z within the context of the associated loss function in Formula 4. Our initial concept was grounded in the understanding that when using the treatment label T as the feature to predict outcome Y, the information in Z might become redundant. Therefore, we sought to disentangle Z from X by minimizing the KL-divergence between the predicted distribution of Y conditioned on both Rz and T, and the predicted distribution of Y conditioned solely on T, i.e., ${D}_{KL}\left[ p(Y \mid {Z},{T}) \Vert p(Y \mid T) \right]$. This served as the motivation for the design of the loss function used in subsequent experiments, and empirical studies have demonstrated its effectiveness. However, we recognize the inadequacy of the previous explanation. Therefore, in the updated version, we have removed the misleading description of the conditional mutual information $I (Z; Y|T)$. We appreciate your insightful feedback!
>
> - Eq 4: In equation (4), we aim to minimize the KL-divergence between the predicted distribution of Y from $R_z$, T and the predicted distribution of Y from T, i.e., ${D}_{KL}\left[ p(Y \mid {Z},{T}) \Vert p(Y \mid T) \right]=-\int p(Y \mid {Z},{T})log \frac{p(Y \mid {Z},{T})}{p(Y \mid T)} dY$. In the revised version, we have removed the relevant statements about $I (Z; Y | T)$. The reasons are detailed in the response to Q1.
>
> - During the training phase, to obtain reasonable representations for C and A, we set up two separate prediction networks from R_c and R_a to Y, which means we use Y as supervisory information to guide the training process of R_c and R_a. This is the only information shared between R_c and R_a during the training process. Therefore, the mutual information between R_a and R_c is all related to Y, and the information contained in Ra that is related to Rc but unrelated to Y is 0 at this stage.
>
> - Corollary 4.3: Thanks for your advice. Here, the notation used in Equation 10 and 11 refers to minimizing all three KL-Divergence together, and we have added related statements in Corollary 4.3 based on your and Reviewer Yekp’s suggestions for better clarity.
>
> - Sec 4.2: We acknowledge that the definitions for these networks need to be sufficiently clear. Below is a brief explanation of these modules, which we have updated in Sec 4.2. **“Retain network”** here represents the neural network for retaining the information from both Z and C. **“Deep networks”** and **“Shallow networks”** are named based on their relative proximity to $R_a$ and $R_c$. Essentially, both are classifiers (for binary scenarios) or regressors (for continuous scenarios).
>
> - Eq. 12: Thank you for your correction. $W$ represents the model weight parameters involved in the $L_2$ regularization loss. As you comprehend, $\alpha, \beta, \gamma$ are global scalars, and $w$ should be example-wise weights. Thus, we modify Eq. 12 in the revised version.
>
> - Metric for “contribution of actual variables”: Following [2] and [3], we use the first(second) slice to denote the weight matrix that connects the variables in X belonging (not belonging) to the actual variables. The polygons’ radii in Figure 3b are scaled between 0: 0.09 and quantify the average weights of the first slice (in red) and the second slice (in blue). For better readability, we have added the description for this metric in Section 5.3.2.
>
> - top of p5: Thank you for pointing out our mistakes. The correct statement is, indeed, "C is independent of A and Z." We have corrected this statement in the new version.
>
> - Maybe consider using m_k etc: Thank you for your suggestion. We have made corrections based on your advice.
>
> - More information of DRCFR: Thank you for your suggestion. We have added relevant statement “ a baseline that ensures the disentanglement of A by proposing Prop 3.1, yet does not provide disentangling solutions for Z and C” in Section 5.3.2 in the new version.
>
>
> **Questions - Response:**
>
> Your understanding is correct. As illustrated in Figure 1, T and Y are causally downstream components of different underlying factors in X in the context of our study.
>
> We share the same assumptions for identifying the treatment effects with [1]. Specifically, suppose the generated IVs satisfy three conditions (Relevance, Exclusion, Unconfoundedness) of valid IVs, additional assumptions are necessary to identify the average causal effect (ACE) of T on Y. Adequate assumptions encompass homogeneity in the causal impact of T on Y, uniformity in the relationship between Z and T, and the absence of effect modification. On the basis of [1], we did not introduce additional conditions for identifiability, as our primary contribution lies in designing an algorithm to identify instrumental variables that satisfy the given conditions. We have also incorporated the description regarding identifiability into Section 3 of the revised paper.

---

> > ### Author Response · Authors · 2023-11-20
> > **References**
> >
> > [1]. Fernando Pires Hartwig, Linbo Wang, George Davey Smith, and Neil Martin Davies. Average causal effect estimation via instrumental variables: the no simultaneous heterogeneity assumption. Epidemiology, 34(3):325–332, 2023.
> > [2]. Negar Hassanpour and Russell Greiner. Learning disentangled representations for counterfactual regression. In International Conference on Learning Representations, 2019.
> > [3]. Learning decomposed representation for counterfactual inference. arXiv preprint arXiv:2006.07040, 2020.

---

> > > ### Comment · Reviewer_pgv6 · 2023-11-22
> > > **Response**
> > >
> > > Thanks for the clarifications. I'll consider raising my score in conversation with other reviewers.

---

> > > > ### Author Response · Authors · 2023-11-23
> > > >
> > > > Dear reviewer, thank you for considering a re-evaluation of your score and for discussing it further with other reviewers. We appreciate the opportunity to clarify any aspects of our work and are ready to provide any additional information needed.

---

### Official Review · Reviewer_FUp5 · 2023-11-05

**Soundness:** 1 poor
**Presentation:** 2 fair
**Contribution:** 2 fair
**Rating:** 3
**Confidence:** 2

**Summary:**

This paper aims to develop an estimator that estimates causal effects in the presence of unobserved confounding while avoid the need to explicitly assume knowledge of an instrumental variable. Instead, they aim to disentangle instruments from confounders and then estimate the effects. They derive an information theoretic approach to separating instruments from confounders and find that it give strong performance on the benchmarks that they tested.

**Strengths:**

* I really like the idea of finding ways to go beyond assuming access to IVs to address unobserved confounding.
 * The method offers strong performance on benchmark datasets.
 * The method makes no distributional assumptions, so if correct, this would make it far more general than anything that has come before it... unfortunately I don't think that it is correct (see counter examples below)

**Weaknesses:**

The method works by minimizing the conditional mutual information between instruments and response given the treatment, $I(Z;Y | T)$, under the claim that the *exclusion* assumption implies that $I(Z;Y | T) = 0$. Unfortunately, this claim is incorrect, because it ignores that conditioning on $T$ opens a collider between $Z$ and $Y$ via $U$ (also via $C$, but the authors are aware of this). Here is an explicit counter example showing that the mutual information is not zero in general:

$ U \sim Bern(0.5 )$

$ Z \sim Bern(0.5 )$

$T = XOR(Z, U)$

$Y = XOR(T,U)$

where $XOR$ is the exclusive or function that evaluates to 1 if either argument is 1 but not both. Notice, that $H(Y | T)$ is just 1, since all the randomness in $Y$ comes from $U$ which has entropy $1$ by construction. More importantly, if you work through all the possible outcomes of the above system, you will see that, conditional on $T$, $Z$ is always equal to $Y$ (i.e. $H(Y | T, Z) = 0$), but that they are conditionally independent when you condition on both $T$ and $U$. For example, if $T = 1$, then when $z=0$ we know $U=1$ (since otherwise $T$ would not be 1), and hence $Y = XOR(1, 1) = 0 = z$.

This is an extreme example, but it highlights that the claim is surely not true in general. Also - the authors make no assumptions on the functional form of the structural equation that determines $Y$, and we know from the LATE (local average treatment effects) framework and Pearl's work on bounding treatment effects, that it is impossible to non parametrically identify the treatment effect with access to an instrument. So there are surely more necessary assumptions for the method to work in general.

The simulation results are very strong, so I don't know whether there is just a missing assumption and the method works under additional assumptions, or if there are mistakes in the simulations too.

**Questions:**

Can you give sufficient conditions under which this method identifies the treatment effect?

---

> ### Author Response · Authors · 2023-11-20
> **Response to Weaknesses and Questions**
>
> **Weaknesses-Response:**
>
> Firstly, we acknowledge there is an issue with the interpretation of the disentanglement process for variable Z within the context of the associated loss function in Formula 4. Our initial concept was grounded in the understanding that when using the treatment label T as the feature to predict outcome Y, the information in Z might become redundant. Therefore, we sought to disentangle Z from X by minimizing the KL-divergence between the predicted distribution of Y conditioned on both Rz and T, and the predicted distribution of Y conditioned solely on T. This served as the motivation for the design of the loss function used in subsequent experiments, and empirical studies have demonstrated its effectiveness. However, we recognize the inadequacy of the previous explanation. Therefore, in the updated version, we have removed the misleading description of the conditional mutual information $I (Z; Y|T)$. We appreciate your insightful feedback!
>
> **Questions-Response:**
>
> We share the same assumptions for identifying the treatment effects with [1]. Specifically, suppose the generated IVs satisfy three conditions (Relevance, Exclusion, Unconfoundedness) of valid IVs, additional assumptions are necessary to identify the average causal effect (ACE) of T on Y. Adequate assumptions encompass homogeneity in the causal impact of T on Y, uniformity in the relationship between Z and T, and the absence of effect modification. On the basis of [1], we did not introduce additional conditions for identifiability, as our primary contribution lies in designing an algorithm to identify instrumental variables that satisfy the given conditions. We have also incorporated the description regarding identifiability into Section 3 of the revised paper.
>
> [1]. Fernando Pires Hartwig, Linbo Wang, George Davey Smith, and Neil Martin Davies. Average causal effect estimation via instrumental variables: the no simultaneous heterogeneity assumption. Epidemiology, 34(3):325–332, 2023.

---

> > ### Comment · Reviewer_FUp5 · 2023-11-21
> >
> > Hartwig et al [2023] assume knowledge of which variable is the instrument. Without that knowledge, their results do not apply - and you are assuming that you do not know which variable in $X$ is the instrument. You can only apply Hartwig et al's assumptions if your method successfully identifies the instrument and as I showed in the counter example, your method will not recover the instrument in general. So you need additional assumptions that explain when it will succeed.

---

> ### Author Response · Authors · 2023-11-22
>
> Thanks for your comment! To ensure the disentanglement of Z, we minimize the KL-divergence between the predicted distribution of Y conditioned on both Rz and T, and the predicted distribution of Y conditioned solely on T, i.e., ${D}_{KL}\left [ p (Y \mid {Z}, {T}) \Vert p (Y \mid T) \right] $. Our initial concept was grounded in the understanding that when using the treatment label T as the feature to predict outcome Y, the information in Z might become redundant. In addition, we set up the representation network of Z and make it align with the causal structure depicted in Figure 1(a) to ensure the generated IVs satisfy three conditions of valid IVs. Identifiability of disentangled representation is challenging, but empirical progress has been made in this direction, for example, [2]. Our method further advances the disentanglement performance. The experimental results in the main paper, Appendix validate the effectiveness of our generated IVs. Thus, following [1], the only assumption in our work is that we can recover the valid IVs from observed features.
>
> [1] Louizos, C.; Shalit, U.; Mooij, J. M.; Sontag, D.; Zemel, R.; and Welling, M. 2017. Causal effect inference with deep
> latent-variable models. Advances in neural information processing systems, 30.
>
> [2] Wu, A.; Kuang, K.; Xiong, R.; Zhu, M.; Liu, Y.; Li, B.; Liu, F.; Wang, Z.; and Wu, F. 2023. Learning Instrumental
> Variable from Data Fusion for Treatment Effect Estimation. In Proceedings of the AAAI Conference on Artificial Intelligence, volume 37, 10324–10332.

---

### Meta-Review · Area_Chair_oxzs · 2023-12-06

**Metareview:**

The paper proposes an approach called the Self-Distilled Disentanglement framework, demonstrating its effectiveness in addressing limitations of mutual information minimization (MIM).

pros:
- The paper works on an important problem of counterfactual prediction with unobserved confounders.

cons:
- there were some concerns about the correctness of the method, and whether the current assumptions are sufficient for causal identification
- the implementation choices in the algorithm lacks sufficient explanation.

**Justification For Why Not Higher Score:**

concerns around the correctness of the method

**Justification For Why Not Lower Score:**

NA

---

### Decision · Program_Chairs · 2024-01-16

Reject